Diagnostic accuracy of Onen’s Alternative Grading System combined with Doppler evaluation of ureteral jets as an alternative in the diagnosis of obstructive hydronephrosis in children

de Bessa Jr Jose bessa@uefs.br 1
Rodrigues Cicilia M. 1
Chammas Maria Cristina 2
Miranda Eduardo P. 3
Gomes Cristiano M. 4
Moscardi Paulo R. 4
Bessa Marcia C. 5
Molina Carlos A. 1
Tiraboschi Ricardo B. 1
Netto Jose M. 6
Denes Francisco T. 4
1 Division of Urology/Public Health, Medical School, Universidade Estadual de Feira de Santana , Feira de Santana , BA , Brazil
2 Department of Radiology, Medical School, Universidade de São Paulo , São Paulo , SP , Brazil
3 Division of Urology, Medical School, Universidade Federal do Ceará , Fortaleza , CE , Brazil
4 Division of Urology, Medical School, Universidade de São Paulo , São Paulo , SP , Brazil
5 Division of Pediatrics, Medical School, Universidade Estadual de Feira de Santana , Feira de Santana , BA , Brazil
6 Division of Urology, Hospital e Maternidade Therezinha de Jesus of the School of Medical Science and Health of Juiz de Fora (HMTJ-SUPREMA), Universidade Federal de Juiz de Fora , Juiz de Fora , MG , Brazil
Menini Stefano
Electronic publication date: 2018 May 18
Publication date: 2018
Volume: 6
Electronic Location ID: e4791
Received 2018 Jan 24; Accepted 2018 Apr 27
Copyright: ©2018 De Bessa Jr et al.
Copyright year: 2018
Copyright holder: De Bessa Jr et al.
License: This is an open access article distributed under the terms of the Creative Commons Attribution License, which permits unrestricted use, distribution, reproduction and adaptation in any medium and for any purpose provided that it is properly attributed. For attribution, the original author(s), title, publication source (PeerJ) and either DOI or URL of the article must be cited.
License URL: https://creativecommons.org/licenses/by/4.0/

Keywords: Hydronephrosis, Urinary tract obstruction, Doppler ultrasound, Diuretic renogram, Children, Diagnostic test, Ureteral jets, Grading system

Funding: The authors received no funding for this work.

==============================
Introduction

Ureteropelvic junction obstruction (UPJO) is a common congenital anomaly leading to varying degrees of hydronephrosis (HN), ranging from no apparent effect on the renal function to atrophy. Evaluation of these children is based on Diuretic Renal Scintigraphy (DRS) and Ultrasonography (US). Recent studies have suggested that new parameters of conventional and color Doppler ultrasonography (CDUS) may be useful in discriminating which kidneys are obstructed. The present study aims to assess the diagnostic accuracy of such parameters in the diagnosis of obstruction in children with UPJO.

Methods

We evaluated 44 patients (33 boys) with a mean age of 6.53 ± 4.39 years diagnosed with unilateral high-grade hydronephrosis (SFU grades 3 and 4). All underwent DRS and index tests (conventional US and CDUS to evaluate ureteral jets frequency) within a maximum interval of two weeks. Hydronephrotic units were reclassified according to the alternative grading system (AGS) proposed by Onen et al. Obstruction in the DRS was defined as a differential renal function <40% on the affected side and/or features indicating poor drainage function like T1/2 >20 minutes after the administration of furosemide, and a plateau or ascending pattern of the excretion curve.

Results

Nineteen hydronephrotic units (43.1%) were obstructed. Some degree of cortical atrophy—grades 3 (segmental) or 4 (diffuse)—was present in those obstructed units. AGS grades had 100% sensitivity, 76% of specificity and 86.4% of accuracy. The absence of ureteral jets had a sensitivity of 73.68%, a specificity of 100% with an accuracy of 88.6%. When we analyzed the two aspects together and considered obstructed the renal units classified as AGS grade 3 or 4 with no jets, sensitivity increased to 78.9%, accuracy to 92%, remaining with a maximum specificity of 100%. These features combined would allow us to avoid performing DRS in 61% of our patients, leaving more invasive tests to inconclusive cases.

Conclusions

Although DRS remains the mainstay to distinguishing obstructive from non-obstructive kidneys, grade of hydronephrosis and frequency of ureteral jets, independently or in combination may be a reliable alternative in the mostly cases.This alternative approach has high accuracy, it is less invasive, easily reproducible and may play a role in the diagnosis of obstruction in pediatric population.

Introduction

Antenatal hydronephrosis (HN) is a common condition seen in 1–5% of all pregnancies. Despite this high incidence seen, only a small percentage of cases will persist and represent a pathology in the postnatal period (Lee et al., 2006). The most common clinically etiology of hydronephrosis is ureteropelvic junction obstruction (UPJO), accounting for more than 50% of cases of high-grade AHN and represents an important cause of kidney damage in infants (Lee et al., 2006; Vemulakonda, Yiee & Wilcox, 2014). However, it often develops with no apparent clinical significance as it may not lead to renal parenchyma injury and may ultimately undergo spontaneous resolution in most children (Eskild-Jensen et al., 2005; Heinlen et al., 2009; Ozcan, Anderson & Gordon, 2004).

Most cases of HN are diagnosed in the antenatal period, having relatively well defined protocols for evaluation and follow-up (Eskild-Jensen et al., 2005; Lee et al., 2006; Ozcan, Anderson & Gordon, 2004; Ransley et al., 1990). Delayed diagnosis of obstructed HN occurs most often in those who lost follow-up or had a progressive worsening of a lower-grade HN. The best way to evaluate and follow-up these older patients, however, is yet to be established. Most protocols recommend serial monitoring with ultrasound (US) and radionucleide diuretic renal scintigraphy (DRS), to determine the degree of HN and parenchyma damage as well the differential renal function (DRF) (Dhillon, 1998; Gatti et al., 2001; Hafez et al., 2002; Lupton & Testa, 1992; Onen, 2007; Peters, 1995). Children with high-grade HN have a higher risk of progression, and kidney damage (Lee et al., 2006). Surgery is indicated upon worsening of hydronephrosis, lithiasis, recurrent infections or when renal function deteriorates (Csaicsich, Greenbaum & Aufricht, 2004; Gatti et al., 2001; Hafez et al., 2002; Onen, Jayanthi & Koff, 2002).

Unfortunately, DRS is an invasive test that involves ionizing radiation and has to be regularly repeated, which might have some harmful consequences in children. Therefore, many studies have investigated extended uses of US in order to minimize the necessity of a DRS. It has been postulated that semi-automated quantitative analysis of US images in HN kidneys may have a potential clinical utility, reducing up to 62% of DRS (Cerrolaza et al., 2016). Other studies have shown good correlation between sonographic findings such as grade of hydronephrosis and ureteral jets pattern and DRF, suggesting a possible role for such parameters in the long-term monitoring of these patients (De Bessa Jr et al., 2008; Erickson et al., 2007; Shapiro et al., 2008; Tekgul et al., 2012; Walker et al., 2015).

The possibility of using an easily available, non-invasive and low-cost method such as ultrasound as an alternative to DRS is certainly attractive. This could be accomplished with an objective assessment of functional and morphological US parameters. The aim of the present study is to evaluate the diagnostic accuracy of renal US and ureteral jet frequency using Doppler US in the bladder for the diagnosis of obstruction in children older than two years old with high-grade unilateral HN.

Materials and Methods

This study was designed based on the most stringent methodological recommendations of the guideline Standard for Reporting of Diagnostic Accuracy (Bossuyt et al., 2003). Eligible patients were two or more years of age and had hydronephrosis grade 3 or 4 according to the Society for Fetal Urology (SFU) grading system (Fernbach, Maizels & Conway, 1993). Grades 3 and 4 are defined by SFU as dilatation of the renal pelvis, and all calyces visible with normal renal parenchyma (grade 3) or atrophic (grade 4). Children with a solitary kidney, bilateral hydronephrosis, neurogenic bladder, hydroureteronephrosis and vesicoureteral reflux were excluded.

This study was approved by the Institutional Review Board of our hospital (protocol number 166/04). Patients agreed to participate after full disclosure of its purposes, and written consent was obtained from all participants or their legal representatives.

Data were collected between October 2006 and June 2014. All patients underwent kidney US, color Doppler ultrasonography to evaluate ureteral jets frequency (UJF)—Index Test—as well as DRS. All tests were performed within a maximum interval of 2 weeks by independent examiners, experienced in clinical research, who were blinded to the results of the other method.

The reports and images were reviewed and cases were reclassified according to the alternative grading system classification (AGS) proposed by Onen et al. (Onen, 2007)—Fig. 1.

Figure 1 Comparison of SFU and Onen’s hydronephrosis grading system AGS.

We used previously described criteria for correlation between the two systems with SFU-1 and SFU-2 hydronephrosis considered as AGS-1 since both present renal pelvic dilatations alone or associated with minimal calyceal dilatation. SFU-3 with calyceal dilation was considered as AGS-2 and SFU-4 classified as either AGS-3 (when renal parenchymal thickness represented half or less of the contralateral normal kidney) or AGS-4 if there was severe parenchymal loss (cyst-like kidney) (Onen, 2007).

Ureteral jets on bladder ultrasound

The technique for evaluation of ureteral jets has been previously described (De Bessa Jr et al., 2008). Subjects were evaluated in dorsal-horizontal decubitus. Employing color Doppler mode, the bladder was examined in the transverse section at the level of the trigone to observe both ureteral orifices simultaneously. The jets were counted during five minutes with the ureteral jets frequency (UFJ) on the hydronephrotic unit side as well as the normal side considered for the subsequent analysis. Relative jet frequency (RJF) was established and defined as the ratio between the UJF in the affected side and the sum of the frequency of jets observed bilaterally in the same period. We considered obstructed, kidneys in which no jets were observed in the hydronephrotic side (UFJ = 0).

Diuretic renal scintigraphy

DRS was performed with 99mTc-DTPA at a dose of 70 to 120 MBq according to the body surface area of the child. Patients were positioned supine and received intravenous hydration (10–15 ml/Kg saline) before the injection of the radiotracer.

Image acquisition began immediately after administration of the radiotracer. In the early stage of flow, for 1 min, images were obtained every 2 s. Between the first and third minutes, one image was obtained every 5 s, and after that the dynamic sequence consisted of one image per minute until the end of the exam. The DRF, expressed as the percentage of each kidney function about overall renal function was calculated according to the relative drug uptake at the initial phase.

Furosemide at the dose of 0.5 to 1.0 mg/kg was injected 20 min after the initial administration of 99mTc-DTPA, to stimulate diuresis. T12 was considered as the time (in minutes) required to excrete half of the renal pelvis radiotracer measured at the moment of furosemide injection.

We considered obstruction when there were signs of kidney damage including impaired differential renal function <40% and/or features indicating poor drainage function like T1/2 >20 min after the administration of furosemide, and a plateau or ascending pattern of the excretion curve (Fernbach, Maizels & Conway, 1993).

Statistical analysis

Measurement of sensitivity, specificity, accuracy, positive and negative predictive values, as well as positive and negative Likelihood ratios are described with their respective confidence intervals of 95%. Kruskal-Wallis test was used to compare DRF between groups. All tests were two-sided and p values less than 0.05 (p < 0.05) were considered statistically significant. The analyses were performed using a statistical software (GraphPad Prism, version 7.0.3; GraphPad Software, San Diego, CA, USA).

Results

Of 46 patients who were eligible for the study, two (4.3%) refused to participate. Forty-four subjects were included with a mean age of 6.5 ± 4.4 years (range two to 17) and a median age of 4.9 years. Of the patients, 33 were boys (75.0%) and 11 were girls (25.0%). The left kidney was affected in 29 (65.9%) children and the right in 15 (34.1%).

Based on the US and UJF (index test), 25 (56.8%) kidneys were classified as AGS-3 or AGS-4, of these, 14 (56.0%) had no detectable ureteral jets. DRS evaluation (reference test) in the same group of 25 children was consistent with significant obstruction associated DRF impairment (DRF <40%) in 19 (76.0%) kidneys, including eight with AGS-4 and 11 with AGS-3. No ureteral jets were detected in 8 kidneys with HN AGS-4 and in 6 of the AGS-3 kidneys. Ureteral jets were present in all of the 19 kidneys classified as AGS-2, and none had abnormal excretion curve nor DRF impairment on DR (Fig. 2). In other words, 19 (43.2%) patients had significant obstruction kidney including 8 AGS-4 and 11 AGS-3, while 25 (56.8%) were unobstructed, including 19 AGS-2 and 6 AGS-3 (Fig. 2).

Figure 2 Flowchart of study design and test results.

In our sample, all 27 patients with obstructive features on DR had DRF impairment. The remaining 17 cases had DRF >40% and were considered as non-obstructive HN as the radiotracer curve of excretion were descendant with T12 less than 20 min.

When DRF was compared between the groups according to the HN classification proposed by ONEN (AGS) a significant difference was found (p < 0.001). Figure 3 demonstrates that all units classified as AGS-4 showed signs of kidney damage while 19 AGS-2 units had preserved renal function. Of the 17 units classified as Grade 3, 11 (64.7%) had renal impairment, and the other six (35.3%) had a normal renal function (Fig. 3).

Figure 3 Differential renal function according to Onen’s AGS.

Measures of diagnostic accuracy, positive and negative predictive values and overall accuracy of conventional ultrasound and UJF on Doppler US in the diagnosis of obstructive hydronephrosis are described in Table 1.

Table 1 Measurements of diagnostic accuracy.

	Sensitivity (95% CI)	Specificity (95% CI)	PPV (95% CI)	NPV (95% CI)	Accuracy (95% CI)	
AGS 3 / 4*	100% [76–100]	76% [55–90.]	76% [55–90]	100% [82–100]	86.4% [76–96]	
AGS 4*	42% [20–66.]	100% [86–100]	100% [76–100]	69.4% [51–83]	75% [62–87]	
Absence of Ureteral Jets*	73.6% [53–93]	100% [76–100]	100% [76–100]	83.3 [65–94]	88.6% [79–98]	
AGS 3 and 4 without jets*	78.9% [60–97]	100% [76–100]	100% [76–100]	86.2% [68–96]	90.9% [83–99]	
Notes.

* p < 0.001.

PPV Positive Predictive Value

NPV Negative Predictive Value

95% CI Confidence Interval 95%

Discussion

As DRS is a challenging diagnostic method with possible negative impact in children following repeated testing, there is a search for less invasive and reliable methods capable of identifying significant obstruction in a timely fashion. The cornerstone of a diagnostic test to these patients should be to discriminate cases more likely to require surgical intervention before renal function deteriorating, from those with innocuous HN. The present study has demonstrated that the combination of morphological and functional parameters of US can safely distinguish obstructed kidneys in children older than two years old.

In our sample, we observed a predominance of HN in males subjects. Moreover, the left side was more affected than right. These findings are consistent with other series in the literature (Fernbach, Maizels & Conway, 1993; Onen, 2007; Palmer et al., 1998; Peters, 1995). We had a higher incidence of high-grade HN in comparison to newborns. This fact has also been reported by Sibai et al. (2001) who reported a higher proportion of HN units with DRF <40% in this subset of patients. Apparently, this population is different from most studied groups whose diagnosis is made prenatally. The majority of prenatal HN resolves in a mean time of 30 months, and those that require intervention will usually be operated in mean age of five months (Ulman, Jayanthi & Koff, 2000).

In the present study, the prevalence of obstructive HN in children older than two years was 43.6%, which is higher than the reported range of 15 to 25% (Eskild-Jensen et al., 2005; Palmer et al., 1998). The mean age of the children in this study was 6.53 years old (range two to 17). This delayed diagnosis might be an effect of asymptomatic individuals who have failed to be identified during prenatal care and incidentally found HN on the abdominal US done for another reason.

We observed that patients with segmental cortical atrophy (AGS-3) had better DRF than diffuse cortical atrophy (AGS-4), proving that this difference is relevant to the chosen classification system for children after their second year of life (Onen, 2007; Sibai et al., 2001). In other words, dividing HN SFU grade 4 into AGS grades 3 and 4, may provide valuable information in the follow-up of high-grade hydronephrosis. Also, T12 has a limited utility in certain age groups, as it has been demonstrated that this parameter might be abnormal in younger children without major obstruction to urine outflow (Ulman, Jayanthi & Koff, 2000). It is an extremely sensitive test but has a high percentage of false positive, which limits its applicability for defining obstruction. However, it has tremendous value in ruling out obstruction, specifically when T12<20 min and curves display with good drainage pattern.

Analyzing US morphological properties, we have found that all units considered obstructed had some degree of cortical atrophy (segmental or diffuse). In contrast, all units that had no cortical atrophy and were classified as AGS-2 or SFU-3 were not obstructed and had DRF >40%. According to our findings, the presence of atrophy (AGS-3 or 4 or SFU-4) is a parameter with 100% sensitivity, specificity of 76% and positive probability index (IP+) of 4.2. Furthermore, the presence of diffuse cortical atrophy (AGS-4) alone had a specificity of 100%, meaning that it consistently predicts kidney damage due to obstruction when present. Similarly, two other studies demonstrated that sonographic findings can be useful to predict deterioration in kidney function in prenatally hydronephrosis. Scalabre et al. (2017) also demonstrated that abnormal parenchymal thickness was a significant predictor of surgical intervention in newborns with prenatally diagnosed unilateral urinary tract dilation. Moreover, Nguyen et al. (2014) (UTD classification system) showed that the anteroposterior diameter of the renal pelvis has similar predictive properties to Onen’s classification. Considering parenchymal atrophy, SFU and UTD system are similar, differing from AGS classification that stratifies it in diffuse or focal atrophy, which may be more precise as we demonstrated herein and described above. Patients with diffuse atrophy had a 100% specificity while those with atrophy not specified (diffuse or focal) had only 76% specificity regarding obstruction. We believe that both systems add value to the original SFU classification. However, further studies are required to define valid cutoffs for pelvic diameters in children older than two years.

Ultimately, during initial US assessment, we could reliably conclude that kidneys with HN and diffuse cortical atrophy (AGS-4) are obstructed, and those with no atrophy (AGS-2) are non-obstructed. Sequential tests would be then required for AGS-3 patients. Recently, Onen’s hydronephrosis grading system has been updated. Parameters such as cortical parenchyma less than 3 mm, the disappearance of corticomedullary differentiation, the absence of medullary parenchyma, and significant hyperechogenicity also define AGS grade 4, which are by our findings (Onen, 2016). Likewise, analysis of UJF has yielded specificity of 100% and sensitivity of 73%, with an overall accuracy of 88.6%. Thus, the absence of jets confirms that obstruction is present while the presence of any jet does not rule out obstruction, given the possibility of occasionally scattered jets less frequently than the contralateral unobstructed unit. We have previously demonstrated that despite higher specificity found in the absence of ureteral jets (RJF = 0), the RJF values that yielded the best accuracy were <25% (De Bessa Jr et al., 2008). However, we believe that complete absence of jets in the hydronephrotic side is potentially more important in clinical decision-making as it simplifies the test and practically confirms obstruction. Finally, a combination of both US parameters (morphological and functional), that is AGS-3/4 and UJF = 0 provided a specificity of 100% and increased the sensitivity to 78.9% and accuracy to 90.9%.

DRS is still considered the gold standard for the evaluation of HN. It has a significant role in defining obstruction, confirming indications for surgical treatment and postoperative monitoring. However, the present study allows us to suggest that the association of renal US and analysis UJF may play a role in the diagnosis of obstruction in children >2 years of age with HN. It has numerous advantages since it is more accessible, non-invasive, with lower cost, does not involve radiation and provides excellent diagnostic accuracy. These characteristics qualify the use of this method in the initial evaluation of hydronephrosis, adding another tool for identification and screening of obstructed and non-obstructed units. In other words, we could have avoided performing DRS in up to 61% of our cases, represented by the spectral extremes of the series (AGS-2 with present UJ or AGS-4 with absent UJ). Future prospective and multicenter studies are necessary to better define the diagnostic role and possible prognostic value of these parameters in the evaluation of hydronephrosis in the neonatal population. Further studies are also necessary to better understand the role of ureteral jets in evaluating post-operative results and need for reintervention.

Conclusions

Although DRS is still the gold standard for the diagnosis of obstructive HN, the association of HN grading and absence of ureteral jets (independently or combined) on color Doppler US is an accurate and easily reproducible option in this scenario. These tests have also demonstrated high sensitivity and specificity, which may help in the diagnosis of obstruction in older children with HN.

Supplemental Information

Supplemental Information 1 Original dataset

Click here for additional data file.

The authors would thank Dr. Abdurrahman Onen for permitting the use of the illustration of his hydronephrosis grading system in this study.

Abbreviations

99m Tc-DTPA Technetium-99m Diethylene Triamine Pentaacetic Acid

AGS Onen’s Alternative Grading System

DRF Differential Renal Function

FUJ Frequency of Ureteral Jets

HN Hydronephrosis

RJF Relative Jet Frequency

SFU Society for Fetal Urology

UTD Urinary Tract Dilation

Additional Information and Declarations

Competing Interests

Author Contributions

Human Ethics

Data Availability

The authors declare there are no competing interests.

Jose de Bessa Jr conceived and designed the experiments, performed the experiments, analyzed the data, contributed reagents/materials/analysis tools, prepared figures and/or tables, authored or reviewed drafts of the paper, approved the final draft.

Cicilia M. Rodrigues performed the experiments, contributed reagents/materials/analysis tools, authored or reviewed drafts of the paper, approved the final draft.

Maria Cristina Chammas conceived and designed the experiments, contributed reagents/materials/analysis tools, authored or reviewed drafts of the paper.

Eduardo P. Miranda analyzed the data, contributed reagents/materials/analysis tools, prepared figures and/or tables, authored or reviewed drafts of the paper, approved the final draft.

Cristiano M. Gomes and Carlos A. Molina conceived and designed the experiments, contributed reagents/materials/analysis tools, authored or reviewed drafts of the paper, approved the final draft.

Paulo R. Moscardi contributed reagents/materials/analysis tools, prepared figures and/or tables, authored or reviewed drafts of the paper, approved the final draft.

Marcia C. Bessa performed the experiments, analyzed the data, contributed reagents/materials/analysis tools, authored or reviewed drafts of the paper, approved the final draft.

Ricardo B. Tiraboschi and Jose M. Netto performed the experiments, contributed reagents/materials/analysis tools, prepared figures and/or tables, authored or reviewed drafts of the paper, approved the final draft.

Francisco T. Denes conceived and designed the experiments, performed the experiments, contributed reagents/materials/analysis tools, authored or reviewed drafts of the paper.

The following information was supplied relating to ethical approvals (i.e., approving body and any reference numbers):

This study was approved by the Institutional Review Board of CAPPesq (protocol number 166/04).

The following information was supplied regarding data availability:

The raw data are uploaded as a Supplemental File.

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
