# Peer review of "Diagnostic accuracy of Onen’s Alternative Grading System combined with Doppler evaluation of ureteral jets as an alternative in the diagnosis of obstructive hydronephrosis in children"

_PeerJ, doi:10.7717/peerj.4791_

## Round 0.1 · original submission · Major Revisions

Dear Drs, Bessa Jr and Miranda,

Your paper has been seen by the editor and two external expert referees. Both reviewers found the paper to be clear and the findings to be of interest. However, they raised some concerns which need to be considered. If these can be satisfactorily addressed, a revised manuscript is likely to be suitable for publication. I enclose below the comments received that set out a number of points which will need your attention before we can consider the submission further. I would urge you to give these points your careful attention; in particular, to the statistical issue raised by the reviewer 1.

Regards,

Stefano Menini

·

Basic reporting

This article is well written, clear and interesting. This study could have meaningfull influence on clinical care of children with hydronephrosis

Experimental design

The research section is well defined and relevant. The method is correct. Authors should be more carefull with the use of the word obstruction, as its definition is not consensual.

The authors did not tell which statisticall test they used to compare impaired renal function between groups.

Validity of the findings

The findings are interesting and properly exposed.

Additional comments

Thank you for your work. The results of this study suggest that renal scintigraphy could be avoided depending on the US parameters. It is well written and the methodology is appropriate.

Abstract :
Please consider dividing the conclusion in two sentences.

Introduction:
Line 61: I find this sentence somewhat ambigious, as hydronephrosis has a high chance of spontaneous resolution, unlike UPJO. Please specify.
Line 70: Lithiasis and infections should be added as potential indications for surgery.
Line 84: "Older children" is too vague and should be children older than 2 years old. Please precise “unilateral HN” in this sentence.

Material and methods:
Line 89: please update reference 21 with the most recent STARD guidelines (2015)

Statistical analysis:
Statistical test used for calculation of p value should be presented in this section. Which test was used to compare DRF between the groups according to the HN classification ?
Line 152: typos: the left kidney (and not .he left kidney)
Line 158: please add a dot at the end of the sentence.
Line 159: In this sentence, obstruction is used for abnormal excretion curve only, and is different from impaired DRF. This definition of obstruction is not consistent with the same word used in the sentence stating the objective of the study.
Line 164: DRF can be normal despite abnormal excretion curves. Would the author define this as obstruction ? It is convenient that no patient of the study was in this situation.
Line 169 is a repetition of line 161, although referring to another figure.

Discussion:
Line 180: Parameters of US are associated with impaired renal function. It is not proven that US can detect children needing surgery prior to renal function deteriorating.
Line 182: again, older children is too vague and should be precised (> 2 years old)
Line 185: the words “on the other hand” are not needed here.
Line 199: A downside of Onen classification is that discrimination between partial of diffuse renal atrophy is not strictly objective.
Line 209 to 210: The prognosis value of abnormal parenchymal thickness has also been demonstrated before: Scalabre et al. Prognostic Value of Ultrasound Grading Systems in Prenatally Diagnosed Unilateral Urinary Tract Dilatation. J Urol. 2017 Apr;197(4):1144-1149. doi: 10.1016/j.juro.2016.11.103.

Line 223 to 225: The use of both Onen classification and UJF improve overall accuracy in this study. However, the new classification UTD ( Nguyen et al. Multidisciplinary consensus on the classification of prenatal and postnatal urinary tract dilation (UTD classification system) J Pediatr Urol. 2014 Dec;10(6):982-98. doi: 10.1016/j.jpurol.2014.10.002.) and the antero posterior diameter of the renal pelvis have predictive values comparable to Onen classification and are simpler to use. Maybe the authors could precise this point in the discussion, or justify why they choose to use this classification in particular.
Moreover, the Accuracy of Onen score combined with UJF was 90.9% in the present study, which is not superior to the accuracy of the antero-posterior diameter of the renal pelvis or the UTD score in previous studies including younger children. Please comment.
Line 237: A postoperative increase of the UJF score would be another way to confirm the correlation between UJF score and obstruction.

Reviewer 2 ·

Basic reporting

• The manuscript is clear, unambigious, Professional English language used throughout.
• The literature is well referenced, and the references are relevant.
• The structure conforms to PeerJ standards and discipline norm.
• The three figures, table and raw data are relevant and well described.

Experimental design

• The data is original primary research within Scope of the journal.
• The research question has well been defined and meaningful.
• The method has been described with suffucient detail.

Validity of the findings

• There are significant meaningful findings benefit to literature, and it has clearly been stated in the text.
• Data is statistically sound and significant.
• Conclusions are well stated and linked to original research question. There is no speculation and conclusion is limited to supporting results.

Additional comments

• Onen et al (Turkish J of Pediatric Surgery – Çocuk Cerrahisi Dergisi, 2016, V30, pp 55) have shown and strongly support your findings regarding the evidence that diffuse cortical atrophy suggest significant obstruction (Onen AGS-4) and renal damage, while Onen AGS-2 units does not develope renal damage (in Lines 167-169 and Lines 207-209 and Lines 213-215). If you read this review in detail, you will find the upgraded Onen’s hydronephrosis grading system is simple, clear and understandable for all specialists including pediatric radiologists and perinatologists. Moreover, the mentioned review has more scientyphic evidens supporting your findings and results. You may find beneficial to add this review as a reference in your present manuscript. It shows clearly the details of anatomic ultrasound findings can suggest many clues on kidney function. Based on detailed and upgraded Onen’s grading system; cortical parenchyme less than 3 mm, disappearance of corticomedullary differantiation, no visual medullary parenchyme and significant hyperechogenecity suggest Onen-4 hydronephrosis which shows significant renal damage.
• The manuscript is clearly written in professional, unambiguous language. Therefore, the English language is clear enough that an international audience can clearly and easily understand your text.
• Your most important issue is that your article clearly demonstrate the significant advantages of Onen’s Alternative Grading System on showing significant obstruction that need surgery with high sensitivity, specificity and accuracy (Lines 209-212). The next most important item is that your manuscript has clearly shown the significant differecense between Onen AGS-3 and AGS-4, and thus the importance of dividing these two group in different grade (Line 196-200); all of the AGS-4 units do need surgery, while the majority of AGS-3 units do not. The third most important issue is that when DRF was compared between the Onen AGS units a significant difference was found (Line 166-167). The last most important point is that the majority of hydronephrotic children (Onen AGS-1,2 and 4) actualy do not need an invasive nuclear scans since it has many disadvantages and false results (Lines 233-235). Renal scan indicated only in Onen AGS-3 (Line 215).
• There is no a significant weakness, therefore the manuscript might be accepted.

---

## Round 0.2 · Minor Revisions

Dear Drs. Bessa Jr and Miranda,

Thank you for your resubmission. I have now received reports from all reviewers who are generally supportive of publication. However, reviewer 1 suggested minor modifications. Accordingly, I invite you to address the reviewer' s comments and recommendations. In particular, the statistical issue regarding the use of ANOVA test needs to be addressed.

Regards,

Stefano Menini

·

Basic reporting

no comment

Experimental design

no comment

Validity of the findings

The use of ANOVA test is only valid when the distributions of the residuals are normal. The authors did not specified if this point was tested before the ANOVA .

Additional comments

The paper was significantly improved, although i would have liked the UTD grading system to be cited in the discussion section (Nguyen 2014).

A few typos remain:

Line 51: in most cases
Line 71: radionucleide
Line 155: between
Line 228: significant

I also understant that ureteral jets were not recorded postoperatively, but this could be an interesting lead for further studies, and could be written at the end of the discussion.

Thank you for your work.

Reviewer 2 ·

Basic reporting

No comment

Experimental design

No comment

Validity of the findings

No comment

Additional comments

This manuscript appears to be appropriate for publication in PeerJ.

---

## Round 0.3 · accepted · Accept

Dear Drs. Bessa Jr and Miranda,

Thank you for submitting a revised version of your manuscript. I am pleased to inform you that your manuscript is accepted for publication in PeerJ in its current form and will now be forwarded to the product editor for copy editing and publication.

I thank all reviewers for their effort in improving the manuscript and the authors for their cooperation throughout the review process

Best regards,

Stefano Menini

·

Basic reporting

no comment

Experimental design

Thank you for these changes, i indeed think that the Kruskal Wallis test is more appropriate.

Validity of the findings

no comment

Additional comments

I thank the authors for their improvement of the paper. The statisticall analysis is now more valid in my opinion.